# The Relationship between Different Large-Sided Games and Official Matches on Professional Football Players’ Locomotor Intensity

**DOI:** 10.3390/ijerph19074214

**Published:** 2022-04-01

**Authors:** Romualdo Caldeira, Élvio Rúbio Gouveia, Andreas Ihle, Adilson Marques, Filipe Manuel Clemente, Helder Lopes, Ricardo Henriques, Hugo Sarmento

**Affiliations:** 1Department of Physical Education and Sport, University of Madeira, 9020-105 Funchal, Portugal; romualdo.caldeira@gmail.com (R.C.); hlopes@staff.uma.pt (H.L.); 2LARSYS, Interactive Technologies Institute, 9020-105 Funchal, Portugal; 3Center for the Interdisciplinary Study of Gerontology and Vulnerability, University of Geneva, 1205 Geneva, Switzerland; andreas.ilhe@unige.ch; 4Department of Psychology, University of Geneva, 1205 Geneva, Switzerland; 5Swiss National Centre of Competence in Research LIVES—Overcoming Vulnerability: Life Course Perspectives, 1015 Lausanne, Switzerland; 6CIPER, Faculty of Human Kinetics, University of Lisbon, 1499-002 Lisbon, Portugal; adncmpt@gmail.com; 7ISAMB, University of Lisbon, 1499-002 Lisbon, Portugal; 8Escola Superior Desporto e Lazer, Instituto Politécnico de Viana do Castelo, Rua Escola Industrial e Comercial de Nun’Álvares, 4900-347 Viana do Castelo, Portugal; filipe.clemente5@gmail.com; 9Instituto de Telecomunicações, Delegação da Covilhã, 6201-001 Covilha, Portugal; 10Centre for Tourism Research, Development and Innovation, University of Madeira, 9004-509 Funchal, Portugal; 11Marítimo da Madeira—Futebol, SAD, 9020-208 Funchal, Portugal; ricardo.henriquesfut@gmail.com; 12University of Coimbra, Research Unit for Sport and Physical Activity (CIDAF), Faculty of Sport Sciences and Physical Education, 3004-504 Coimbra, Portugal; hg.sarmento@gmail.com

**Keywords:** soccer, large-sided games, total distance, exercise intensity, pitch size

## Abstract

Large-sided games (LSG) are commonly used in the training contexts for providing either technical/tactical or locomotor/physiological stimuli. Despite natural similarities with the official match, the locomotor profile seems to be different, which must be considered by the coaches to identify compensatory strategies for achieving the ideal dose of training. The aim of this study was two-fold: (1) to investigate the locomotor demands imposed by LSGs and the official matches; and (2) to compare the effect of different pitch sizes’ LSG conditions in the locomotor demands. This study followed an observational design. Sixteen professional football players from the same team (26.3 ± 3.0 years old) were included. The study was conducted over four weeks. The same GK + 10 × 10 + GK play format with different pitch sizes (i.e., area per player ranging between 195 m^2^ to 291 m^2^) was analyzed. Three official matches were also collected in which the 10 most demanding minutes were considered for further comparisons. Only the same players who participated in matches were considered in comparison with the LSG. The data were obtained using a 10-Hz global positioning system technology. Total distance (TD) and mechanical work (MW) scores increased 20% and 23%, respectively, between the smallest and biggest pitch sizes (*p* < 0.001). There was a significant difference in locomotor intensity metrics between opponents from different positions on the table (*p* = 0.001). The biggest LSG (i.e., 291 m^2^ per player) was the only one that required similar levels of locomotor intensity as required in the official full match. The present study demonstrates that LSG pitch size variation requires different locomotor intensities. Bigger pitch sizes cause an increase in TD and MW. In addition, considering the position on the table, the level of opponents induces different TD covered. Finally, the largest LSG simulates the official match more accurately.

## 1. Introduction

One of the major challenges in football is building training exercises that allow players to be confronted with real match scenarios while continuously enhancing their performance (e.g., tactical, technical, physical, physiological, psychological). In this sense, small- and large-sided games (LSG), also known as small- and/or large-sided conditioned games, are very popular training resources commonly used by coaches to replicate those scenarios [1,2,3]. The design of sided games, considering the formal game format, allows the chaotic effect of football to be simplified without compromising the fundamental characteristics of the match or its dynamic and complex quality [4]. Besides promoting the strategical-tactical side, they also simulate the physiological/physical part of the game [5].

The configuration of these training exercises is strictly related to manipulating task constraints. For a typical task, constraints could include changes in format (e.g., number of players involved in the game and numerical relationships), scoring method (e.g., having or not having goalkeepers, using or not using goals or targets), strategical-tactical missions (e.g., coach-specific instructions), training regimen (e.g., work-to-rest ratio, sets, repetitions) and/or pitch configuration (e.g., dimensions of the playing area, area per player, the shape of the pitch, width-to-length ratio) [1,6]. The management of these variables is well documented in the literature. In fact, in recent years, the interaction between these task constraints has been widely investigated to modify players’ acute and chronic load responses to levels that elicit physiological match-specific adaptations [7,8].

Nowadays, the association between training and match monitoring has been one of football’s most common research topics [9]. The technological and analytical method evaluations have provided to coaches, members of technical staff and sports scientists several load measures obtained throughout the use of global positioning systems (GPS), among other microtechnologies (e.g., accelerometers, heart rate monitors, etc.) [10]. Within the load measures categorization, the locomotor intensity by means of GPS has been widely used for training load assessment and monitorization. The evidence around these locomotor activities is clear, since the internal load is strongly associated with the amount of running completed rather than the several other locomotor intensity measures typically monitored in team-sport players [11]. The most common metrics given by GPS include distances, accelerations, decelerations, impacts and total load. Within distances, one of the most used locomotor intensity measures in the assessment of the amount of work developed by the players in training and games is total distance covered (TD), which is measured in absolute (m), and relative values (m/min, m/15 min, m/h) as a percentage of the highest data reached in the match [10]. Within the accelerations, mechanical work (MW), is one of the most used variables to describe locomotor intensity. This metric was created to quantify the total load that players are exposed to and is based on the acceleration data recorded by triaxial accelerometers [12] with high reliability and validity [13,14]. It can be measured in absolute (arbitrary units (AU) and g) and relative (AU/ min, g/min and AU/m) values [10] and as a percentage of the highest data reached in the match. Several research works have focused on these two metrics for analyzing the physiological/physical demands placed on professional football players [15,16,17,18,19,20].

Although LSG appears to be a beneficial means to comprehend the players’ performance in official matches, it remains a relatively unpopular topic in the literature [2]. Namely, there is a shortage of consistency in the design of LSGs, the age and level of ability of the players and the pitch size variation used by researchers. Moreover, the generality of research in this area is conducted with young players [21,22], while studies conducted with senior players [23,24,25], and particularly with professional players, are rare. Further, when it comes to applying the optimal constraints to the training task and the appropriate management of players’ training load, match activity plays a key role [26,27]. To the best of our knowledge, only very few studies have explored the relationships between GPS locomotor intensity indicators and official matches, especially with large formats (e.g., GK + 10 vs. 10 + GK) and with the highest data reached in the game regarding key locomotor intensity indicators, such as TD and MW. This novel information will be crucial to identify which LSG conditions, in terms of pitch sizes, provide professional football players with a better level of readiness for competition.

Therefore, this study aimed to compare the locomotor demands imposed by LSG and the official match, and to compare the effect of different pitch sizes in the locomotor demands.

## 2. Materials and Methods

### 2.1. Experimental Approach

The study followed an observational study design. Microelectromechanical systems monitored locomotor demands in three consecutive matches and LSG applied in different training sessions. The observational period occurred over four weeks. The data collection started six weeks after the season began (corresponding to the first half of the season).

The official matches occurred at the weekends. The LSG were monitored over 3 days after the match. The data collection in training sessions occurred between 04:00 and 05:30 p.m. on natural grass field with sunny weather conditions and similar temperatures. In the case of the official matches, the data collection occurred between 06:00 and 10:30 p.m. on natural grass field with sunny weather conditions and similar temperatures.

### 2.2. Participants

A priori sample calculation was determined based on T-Test Family—Wilcoxon signed-rank test (i.e., matched pairs). This indicated that to detect a large effect size of r = 0.90 with an alpha probability of 0.05, a power of 0.95, the sample size would need to comprise 16 participants. GPower, (Heinrich Heine University, Düsseldorf, Germany; 3.1.9.7 software) was used in the calculations [28].

Sixteen male professional outfield football players from the same team (i.e., age: 26.3 ± 3.0 years old; height: 181.6 ± 5.2 cm; body mass index: 23.8 ± 1.5 kg/m^2^; percentage fat mass: 10.4 ± 2.9) participating in the Portuguese premier league were included in this analysis. Of the players, 7 were defenders (DF), 6 were midfielders (MF), and 3 were forwards (FW). The data were collected from daily player locomotor demands in which player activities were routinely measured during training sessions and matches. The following inclusion criteria were used: (1) players were included in the analysis if they participated in all training sessions of the week where the LSG and official match occurred; and (2) players also had to have participated for at least 45 min in one of the official matches, as suggested by previous research [19,29] (Table 1).

The players were informed of the study design and the benefits and consequences of their participation and freely signed an informed consent form. All procedures were approved by the Ethical Committee of Faculty of Human Kinetics, University of Lisbon (CEIFMH n.º: 35/2021) and followed the ethical standards of the Declaration of Helsinki for a study in humans.

### 2.3. Large-Sided Games

The most used LSG played during the period of the study (i.e., GK + 10 vs. 10 + GK) were compared with the 10 most demanding minutes reached in the official matches. Keeping the same format, the LSG ranged in the pitch’s size as follows: (1) Condition 1 (CD1): 67 × 64 m, average area per player = 195 m^2^; (2) CD2: 78 × 68 m, average area per player = 241 m^2^; and (3) CD3: 100 × 64 m, average area per player = 291 m^2^ (Table 2)

All LSG lasted 10 min and integrated a standard training session (e.g., tactical, technical and physical factors were amalgamated) within a typical training week compound of 5 field sessions. The training week consisted of a typical microcycle structure using the following schedule: previous match-day (MD); MD + 1 and MD + 2: recovery period; MD-4 and MD-3: acquisition period; MD-2 and MD-1: tapering period; and next MD.

All the official game rules were maintained for the three LGS, except for the offside rule and the start and restart of play rule. Every time one of the teams won a free kick (direct and indirect), a penalty, a throw-in or a corner kick, the restart of the game was performed by the GK of the team to whom a goal kick belonged. There were four balls always placed next to the goal of the goalkeeper to whom the ball belonged to make this replacement process quick. The verbal encouragement of the coach remained the same throughout the three LGS.

### 2.4. Official Matches

Official match activities were assessed from data collected over three official 11-a-side games. Three different official matches were selected according to opponents’ level. Considering the classification in the championship, three different levels were selected: lower, middle and upper position on the table. Each microcycle throughout the study finished with an official match. The team’s management remained the same throughout the study. The team systematically played in a 1-4-3-3 formation, with 4 DF, 3 MF and 3 FW. The same warm-up protocol was conducted before each official match, including moderate running, dynamic stretching, mobility and balance exercises, accelerations and decelerations, and ball possession drills in a 5 vs. 5 configuration. The average playing area per player across the three official matches was approximately 325 m^2^.

### 2.5. Locomotor Demands 

Each player’s movements were recorded by 10-Hz GPS Unit (EVO, Catapult, Melbourne, Australia) during each training session and official match. The GPS unit also included an accelerometer, a gyroscope and a magnetometer (100 Hz, 3 axes ± 16 g). The GPS device was put in a skin-tight bag in the thoracic region between the scapulae. Data were collected during what was good weather and satellite conditions for GPS training sessions and matches [19]. The following measures were collected during each training session and official match: (1) total distance (TD: consisting of the total distance covered by each player); and (2) mechanical work (MW: total load players are exposed to and is obtained from the acceleration in the three axes recorded by the GPS accelerometers, measured in arbitrary units [AU].

Catapult open field cloud was then used to compute a moving average over each criterion variable (TD and MW) for the official matches, using a 10 min duration, and the maximum value per player was recorded. These data were then averaged for each one of the three competitive matches. The LSG conditions lasted for 10 min and were also analyzed for each player and averaged as three different training drills (corresponding to each pitch dimension used). Descriptive statistics and analysis were then calculated based on this design.

### 2.6. Statistics

First, descriptive statistics were calculated (means, standard deviation and confidence intervals) for age, body composition and physical fitness variables.

Second, the Friedman test was used to identify TD and MW changes across the three LSG formats and the three competition moments.

Third, a Wilcoxon signed-rank test was conducted to analyze individual differences in the scores of the locomotor intensity between each LSG condition and the official full game. Statistical analysis was performed using IBM SPSS Statistics v.26.0 (IBM, Armank, NY, USA). The significance level was set to *p* < 0.05.

## 3. Results

First, descriptive statistics of age, body composition and physical fitness variables are presented in Table 3.

Second, the results of the Friedman test indicated that there was a significant difference in the total distance (TD) scores [χ2 (2, n = 16) = 15.50, *p* < 0.001, Figure 1] and mechanical (MW) scores [χ2 (2, n = 16) = 14.00, *p* = 0.001, Figure 2] between the three different pitch size conditions in GK + 10 vs. 10 + GK. Inspection of the median values showed that the GK + 10 × 10 + GK (291 m^2^) format presented the highest score in TD and MW.

For the three official matches, there was a significant difference in the TD scores [χ2 (2, n = 16) = 13.88, *p* = 0.001]. The match against the best-classified team presented the lowest score in TD. No significant difference was seen for MW [χ2 (2, n = 16) = 1.41, *p* = 0.494]. 

Third, a Wilcoxon signed-rank test was conducted to analyze a change in TD and MW from training conditions and to official matches.

### 3.1. Total Distance

The Wilcoxon signed-rank test revealed a significant difference in TD between GK + 10 vs. 10 + GK (i.e., smallest pitch size) and the Official Match 1 (i.e., against a team with a low position on the table) z = −2.40, *p* = 0.017 (r = 0.60) and Official Match 2 (i.e., against a team with a middle position on the table) z = −2.51, *p* = 0.012 (r = 0.63). The median score on the TD revealed lower scores in the training condition when compared with the official match (Table 4).

There was a significant difference in TD between GK + 10 vs. 10 + GK (i.e., medium pitch size) and the Official Match 1 (i.e., against a team with a low position on the table) z = −3.52, *p* < 0.001 (r = 0.88), Official Match 2 (i.e., against a team with a middle position on the table) z = −3.36, *p* < 0.001 (r = 0.84) and Official Match 3 (i.e., against a team with a upper position on the table) z = −2.64, *p* = 0.008 (r = 0.66). The median score on the TD revealed lower scores in the training condition when compared with all official matches (Table 4).

Finally, there was a statistically significant difference in TD between GK + 10 vs. 10 + GK (i.e., large pitch size) and the Official Match 1 (i.e., against a team with a low position on the table) z = −2.22, *p* = 0.026 (r = 0.56) and Official Match 3 (i.e., against a team with an upper position on the table) z = −2.54, *p* = 0.011 (r = 0.63). The median score on the TD revealed higher scores in the training condition when compared with the official match (Table 4).

Figure 3 depicts the players’ individual variation of locomotor demands (i.e., total distance and mechanical work) across the three official matches and GK + 10 vs. 10 + GK LSG conditions.

### 3.2. Mechanical Work

The Wilcoxon signed-rank test revealed a significant difference in MW between GK + 10 vs. 10 + GK (i.e., smallest pitch size) and the Official Match 1 (i.e., against a team with a low position on the table) z = −3.06, *p* = 0.002 (r = 0.77), Official Match 2 (i.e., against a team with a middle position on the table) z = −2.95, *p* = 0.003 (r = 0.74) and Official Match 3 (i.e., against a team with a upper position on the table) z = −2.69, *p* = 0.007 (r = 0.67). The median score on the MW revealed lower scores in the training condition when compared with all official matches (Table 4). 

There was a significant difference in MW between GK + 10 vs. 10 + GK (i.e., medium pitch size) and the Official Match 1 (i.e., against a team with a low position on the table) z = −3.06, *p* <.001 (r = 0.88), Official Match 2 (i.e., against a team with a middle position on the table) z = −2.95, *p* = 0.001 (r = 0.83) and Official Match 3 (i.e., against a team with a upper position on the table) z = −2.69, *p* = 0.001 (r = 0.80). The median score on the MW revealed lower scores in the training condition when compared with all official matches (Table 4).

Finally, no significant differences in MW were seen between GK + 10 vs. 10 + GK (i.e., large pitch size) and any official match.

## 4. Discussion

The novel findings of the present study showed a difference in locomotor intensity metrics when pitch sizes change in LSG. Bigger pitch sizes caused an increase in TD and MW. There was a significant difference in locomotor intensity metrics between opponents from different positions on the table (i.e., lower, medium or upper). Competition against teams from the upper position caused a lower TD. The biggest LSG (291 m^2^ per player) was the only one that required similar levels of locomotor intensity as required in the official match. This study gives important information for coaches to plan LSG formats with similar locomotor intensity to those they will find in the official match.

First, keeping a similar LSG format, we analyzed the pitch size variation in total distance covered and the MW. A recent umbrella review of systematic reviews and meta-analyses on SSG in team ball sports revealed that pitch configuration (i.e., mainly the pitch size and the influence of different relative areas of play) was currently one of the main game constraints discussed in the literature [8]. However, it is claimed more research should investigate the effect of manipulation of playing field dimensions on the emerged outcomes of the games [30], since some studies differed significantly in their results and interpretations of the findings. In addition, the information about LSG (e.g., GK + 9 vs. 9 + GK; GK + 10 vs. 10 + GK) in professional football is scarce. Our results reinforce the idea that larger pitches cause increases in the TD and the MW. It means that the larger the playing field dimensions during practice, the greater the distances between players in the same team and between those players and the opposing team. Since LSG are widely used by coaches, the present study results have important practical implications and give helpful information to guide methodologies used in professional football training.

Secondly, in this study, we investigated the variation of TD and MW considering different levels of opponents (i.e., low, middle and upper position on the table). Clemente et al. [8] concluded that greater competitive levels increase locomotor intensity during games. The present study contradicts these findings, since it was found that the competition moment against the team from the upper position caused a lower total distance run. Despite the tactical-technical approach of the players that may influence the team’s behavior, it is comprehensive that playing against an opponent from the upper position on the table generates a bigger concern about the cohesion of the defensive process that may cause a huge contention of the team and a lower distance covered.

Third, we compared the different LSG (i.e., in terms of pitch sizes) and the real moment of competition on TD and the MW, considering the levels of the opponents. This information allows us to understand better which LSG more accurately simulate the official full match. Previous studies have compared the relative physical demands of SSG (medium and large) in official matches and in non-professional football players [3,31,32,33]. However, to our knowledge, this is the first study that considers the levels of opponents when comparing the LGS. This opens new and important perspectives about the teams’ preparation for specific matches.

In the present study, independently of the opponents’ position on the table, the largest LGS (i.e., 291 m^2^) simulates the official match more accurately than other sided games in terms of TD and MW metrics. Similar results were seen by authors of Ref [33], who concluded that GK + 9 vs. 9 + GK simulates the official match more accurately than other large-sided and/or conditioned games in sprinting and loading demands. These differences in TD between official matches and large-sided and conditioned games also agree with previous reports [3,32]. It is also important to underline that the largest LSG format required a bigger mean TD covered (i.e., 1273.4 ± 101.9 m) when compared to the official match against a team with a lower position on the table (i.e., 1203.6 ± 74.7 m) and the official match against a team with an upper position on the table (i.e., 1156.5 ± 87.1 m). It means that the largest GK + 10 vs. 10 + GK (i.e., 291 m^2^) was the format studied that better prepared the team for the official match. Based on these results, coaches should consider this format to better simulate the TD and the MW required by the official matches.

We acknowledge that the differences in constraints between large-sided and conditioned games in previous studies make the results difficult. However, it is believed that LSG better simulates the official match in terms of TD and MW, which is an important message for coaches. This is in line with the ecological dynamics approach that supports LSG as a practice methodology that ensures task constraints during training are similarly demanding to a competitive context [34,35]. It means that, probably, LSG forces players in a better way to adapt their overall actions to a changing performance environment, similarly to competitive performance conditions. It is also important to acknowledge that once we are analyzing the players’ performance, the temperatures and relative humidity for each training session and official match should be registered in the future, since hydration status and ambient conditions are important variables that may affect the players’ performance.

## 5. Conclusions

The present study demonstrates that pitch size variation (i.e., 195 m^2^; 241 m^2^; 291 m^2^) in LSG requires different locomotor intensity in professional football players. A bigger pitch size causes an increase in TD and MW. In addition, the level of opponents, considering their position on the table, induces a differently covered TD. The competition moment against teams from the upper position causes a lower TD run. Finally, the largest LSG format more accurately simulates the official full match. These novel results have important implications and give precious detailed information for coaches to better plan LSG formats with similar locomotor intensity to those they will find in the competition moment.

## Figures and Tables

**Figure 1 ijerph-19-04214-f001:**
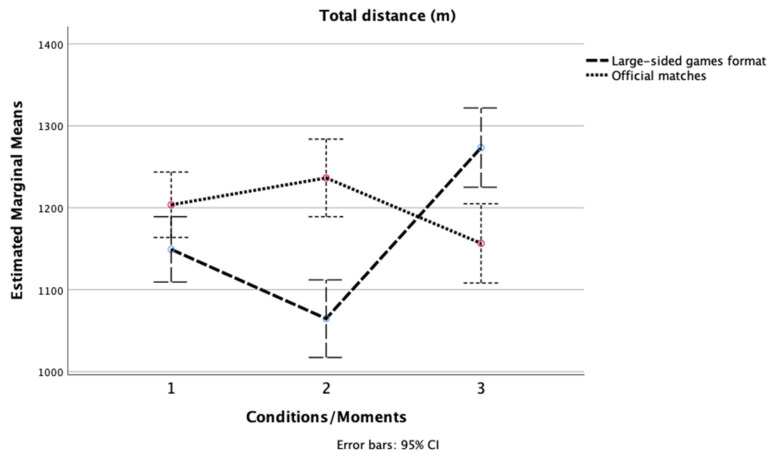
Changes in total distance across the three LSG formats and the three official matches.

**Figure 2 ijerph-19-04214-f002:**
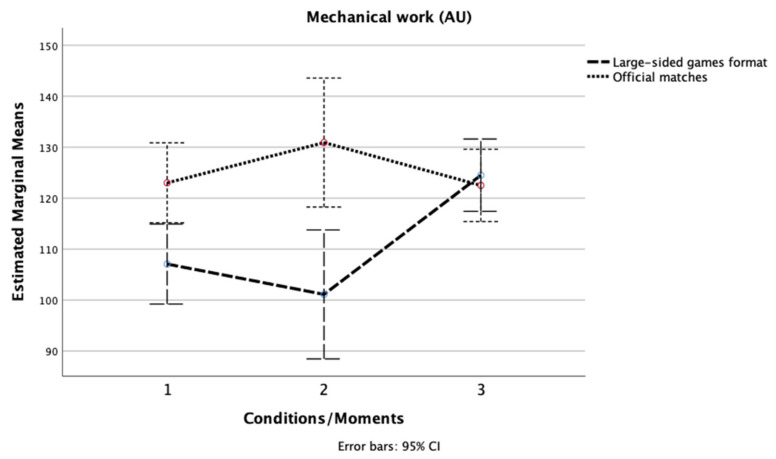
Changes in mechanical work across the three LSG formats and the three official matches.

**Figure 3 ijerph-19-04214-f003:**
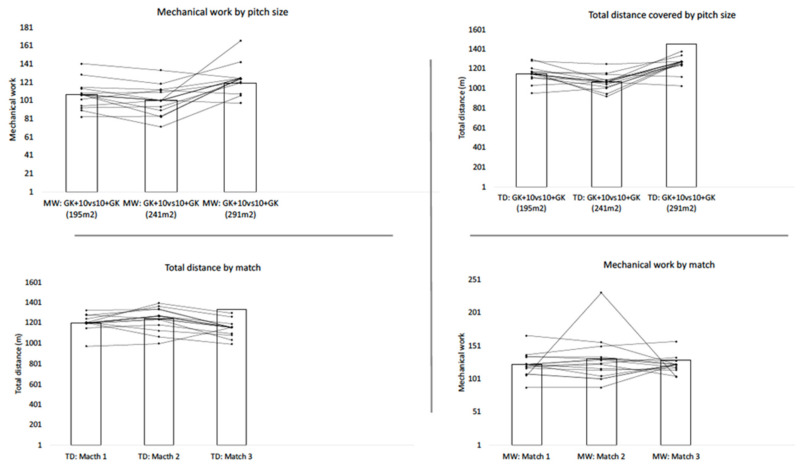
Players’ individual variation of locomotor demands (i.e., total distance and mechanical work) across the three official matches and GK + 10 vs. 10 + GK LSG conditions. The columns represent the average score for the group.

**Table 1 ijerph-19-04214-t001:** Players’ individual descriptive of the official matches.

	Official Match 1	Official Match 2	Official Match 3
Date	31 October 2020	7 November 2020	30 November 2020
Home/away game	Home	Away	Home
Game score	Draw	Loss	Loss
Player 1	100% of the time	100% of the time	100% of the time
Player 2	100% of the time	100% of the time	
Player 3	100% of the time	100% of the time	90% of the time
Player 4	100% of the time	100% of the time	
Player 5	100% of the time	76% of the time	96% of the time
Player 6	100% of the time	48% of the time	
Player 7	82% of the time		
Player 8	60% of the time	48% of the time	
Player 9	100% of the time	84% of the time	100% of the time
Player 10	85% of the time	100% of the time	76% of the time
Player 11		100% of the time	100% of the time
Player 12		51% of the time	100% of the time
Player 13		51% of the time	
Player 14			100% of the time
Player 15			100% of the time
Player 16			76% of the time

**Table 2 ijerph-19-04214-t002:** Large-sided games characteristics.

	Week 1	Week 2	Week 3
Format	GK + 10 vs. 10 + GK	GK + 10 vs. 10 + GK	GK + 10 vs. 10 + GK
Pitch size	67 m × 64 m (4288 m^2^)	78 m × 68 m (5304 m^2^)	100 m × 64 m (6400 m^2^)
Area per player	195 m^2^	241 m^2^	291 m^2^
Task objectives	The main objective for the three LSG conditions was to score as many as goals as possible and not to to give any possible chance for the opponent to score.		
Task rules	All the official game rules were maintained for the three LSG, except for the offside rule and the start and restart of play rule. Every time one of the teams won a free kick (direct and indirect), a penalty, a throw-in or a corner kick, the restart of the game was performed by the GK of the team to whom the goal kick belonged. All LSG were also played with free touch rule per player.		
Sets	1		
Minutes per set	10′		

**Table 3 ijerph-19-04214-t003:** Descriptive statistics of age, body composition and physical fitness variables.

	Mean	SD	95% Confidence Interval for Mean	
			Lower Bound	Upper Bound
Age (years)	26.3	2.9	23.8	28.7
Weight (kg)	78.6	6.2	73.3	84.8
Height (cm)	181.6	5.2	178.7	184.3
BMI (kg/m^2^)	23.8	1.5	22.5	25.3
Percent of body fat	10.4	2.9	8.9	11.9
Countermovement jump (cm)	39.0	4.8	35.3	42.7
Squat jump (cm)	38.1	4.5	34.4	41.6
Handgrip strength (kg)	50.6	8.1	44.9	58.1
Sprint 35 m (s)	4.9	0.3	4.7	5.2

**Table 4 ijerph-19-04214-t004:** Individual differences in the scores of the locomotor intensity between each LSG condition and the official match.

	GK + 10 vs. 10 + GK (195 m^2^)	Official Match 1		Official Match 2		Official Match 3	
	M ± SD; Med	M ± SD; Med	*p*	M ± SD; Med	*p*	M ± SD; Med	*p*
AvgTD	1149.2 ± 81.5; 1149.0	1203.6 ± 74.7; 1203.6	0.017	1236.4 ± 105.2; 1236.4	0.012	1156.5 ± 87.1; 1156.5	0.877
AvgMW	107.1 ± 14.1;107.0	123.0 ± 16.6; 123.0	0.002	130.9 ± 31.7; 130.0	0.003	122.5 ± 12.0; 122.5	0.007
	**GK + 10 vs. 10 + GK (241 m^2^)**						
AvgTD	1064.6 ± 78.4;1064.6	-	<0.001	-	0.001	-	0.008
AvgMW	101.1 ± 15.0; 101.1	-	<0.001	-	0.001	-	0.001
	**GK + 10 vs. 10 + GK (291 m^2^)**						
AvgTD	1273.4 ± 101.9; 1273.4	-	0.026	-	0.196	-	0.011
AvgMW	124.5 ± 15.6; 124.5	-	0.918	-	0.717	-	0.437

AvgTD, mean total distance; AvgMW, mean mechanical work; Official Match 1, against a team with a low position on the table; Official Match 2, against a team with a middle position on the table; Official Match 3, against a team with an upper position on the table. *p* < 0.05; Wilcoxon signed-rank test.

## Data Availability

The data presented in this study are available on request from the corresponding author.

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
