# Peer review of "The Relationship between Different Large-Sided Games and Official Matches on Professional Football Players’ Locomotor Intensity"

_ijerph, 2022, doi:10.3390/ijerph19074214_

Round 1

Reviewer 1 Report

I think that the contribution in the text of the value of the statistic (z and χ2) and the p-value is redundant, makes reading difficult and does not provide additional information. In my opinion I would leave only the p value.

Review the nomenclature of the graphs, the title of figure 1 is not appreciated

The BMI unit is kg/m2 not kg as an index in the text

Author Response

Dear reviewer, 

Thank you for the opportunity to revise our manuscript (IJERPH-1641565) and resubmit it to the International Journal of Environmental Research and Public Health. 

We are very grateful 

for the overall positive evaluation as well as for the detailed and helpful suggestions for improvement. In a thorough revision, we have now addressed all of the comments raised and feel that the manuscript has substantially been improved as a result.

We hope the revised version is now suitable for publication in the International Journal of Environmental Research and Public Health.

Yours sincerely,

  1. I think that the contribution in the text of the value of the statistic (z and χ2) and the p-value is redundant, makes reading difficult and does not provide additional information. In my opinion I would leave only the p value.

Response: We followed the reviewer’s suggestion and cleaned the r value in table 4.

  1. Review the nomenclature of the graphs, the title of figure 1 is not appreciated

Response: Figure 1 and figure 2 were revised accordingly.

  1. The BMI unit is kg/m2 not kg as an index in the text

Response: We consider the reviewer’s suggestion and correct the unit of measurement on the text.

Reviewer 2 Report

The present observational study approaches the analysis of locomotor demands in football, comparing the demand during official matches and small-sided games (typical training medium in football). Authors analyzed and compared data from 3 different size sided games (195, 241 and 291 m2 per player) and a portion of 10 minutes (the most demanding) of three different official matches (325 m2). Authors conclude that larger SSG better simulated the locomotor demands observed in official matches.

This is an interesting study, since exercises and task usually used in football training are sometimes insufficiently studied in order to provide a proper training stimulus according to competition demands. In general, the study is well designed and seemed to be well conducted. However, there are some concerns that I would like to point: 

Line 52: The use of the terms “small-sided games (SSG and SSGs)”, “large-sided games (LSG)” are confusing because authors used them indistinctly. Further, expressions such as “largest SSG” and “largest LSG” contribute to this confusion. Please, specify and unify the terms throughout the document.

Line 54: See? This reference refers to small-sided games. Thus, the use of LSG seems confusing to me.

Lines 77-78: are they really innumerable?

Lines 106-107: clarify that you refer to pitch sizes where SSG are developed.

Lines 112-113: I think you meant “six weeks after the season began”, right?

Lines 114-115: first, change thought for through. Second, were the 3 types of SSG monitored every day, or one each day? How was then the data selected according to it?

Lines 116 and 119: Did you registered the exact temperatures and relative humidity for each training session and match? Please, provide this data if you have them. Also, did you control the hydration status of your participants? Since you analyzed performance, hydration status and ambient conditions are important issues to control in order to ensure that your data was not influenced by dehydration (DOI: 10.1007/s40279-017-0738-7)

Lines 179-185: I am sorry, but I still don't get how you treated your data. How many SSG in how many sessions were monitored for each pitch size? Where they monitored during 10 min or during the whole training session but selected the 10 minutes of maximum effort? This maximum effort was average for the 16 players (all monitored the same time portion) or each player data came from different time portions? Same for the match performance.

Lines 189 and 191: Why did you use non-parametric statistics procedures?

Line 204: I am sorry. This is the first time you indicate that the matches were against different level rivals according to classification. I believe that this should have been explained in the methods section. Also, what was the level of the team of the 16 players? Because depending on that, motivation when playing upper o lower level rivals may be different, and thus, performance.

Lines 207-208: I think you mislead this phrase.

Figure 2: It does not look properly. X axis labels are cut.

Lines 236-238: when you say competition, I think you want to say match. I know it means the same, but it is better to unify terms.

Lines 278 and 280: Again, they may be used indistinctly, but it seems confusing to me when you speak of SSG and LSG as the same concept.

Line 293: Please, indicate the main author of this reference (6) for better reading.

Lines 296-299: I think this affirmation may be logical. However, the conclusion of an umbrella review seems stronger to me. I believe that not always a lower level team plays defensive against an upper level rival. The tactical approach of the staff may influence as well.

Line 363: Please, review the references, some of them do not seem to follow the format of the journal.

Author Response

Dear reviewer,

Thank you for the opportunity to revise our manuscript (IJERPH-1641565) and resubmit it to the International Journal of Environmental Research and Public Health

We are very grateful for the overall positive evaluation as well as for the detailed and helpful suggestions for improvement. In a thorough revision, we have now addressed all of the comments raised and feel that the manuscript has substantially been improved as a result.

We hope the revised version is now suitable for publication in the International Journal of Environmental Research and Public Health.

Yours sincerely,

Author's Reply to the Review Report (Reviewer 2)

The present observational study approaches the analysis of locomotor demands in football, comparing the demand during official matches and small-sided games (typical training medium in football). Authors analyzed and compared data from 3 different size sided games (195, 241 and 291 m2 per player) and a portion of 10 minutes (the most demanding) of three different official matches (325 m2). Authors conclude that larger SSG better simulated the locomotor demands observed in official matches. This is an interesting study, since exercises and task usually used in football training are sometimes insufficiently studied in order to provide a proper training stimulus according to competition demands. In general, the study is well designed and seemed to be well conducted. However, there are some concerns that I would like to point: 

  1. Line 52: The use of the terms “small-sided games (SSG and SSGs)”, “large-sided games (LSG)” are confusing because authors used them indistinctly. Further, expressions such as “largest SSG” and “largest LSG” contribute to this confusion. Please, specify and unify the terms throughout the document.

Response: We consider the reviewer’s suggestion and unify the term for “large-sided games (LSG)” throughout the manuscript.

  1. Line 54: See? This reference refers to small-sided games. Thus, the use of LSG seems confusing to me.

Response: We agree that the statement could be confused, and we correct the phrase in the text.

  1. Lines 77-78: are they really innumerable?

Response: We modified the phrase on the manuscript, using “several” instead of “innumerable”.

  1. Lines 106-107: clarify that you refer to pitch sizes where SSG are developed.

Response: To better clarify the constraints that we investigate in this study, we included “pitch sizes” as follows: “(…) identify which LSG conditions in terms of pitch sizes, provides (…)”.

  1. Lines 112-113: I think you meant “six weeks after the season began”, right?

Response: Yes. This information was added to the text for a better understanding.

  1. Lines 114-115: first, change thought for through. Second, were the 3 types of SSG monitored every day, or one each day? How was then the data selected according to it?

Response: First, we followed the reviewer’s suggestion and changed “thought” for “through”. Second, the 3 types of SSG were monitored on different days. Third, the SSG conditions lasted for 10 minutes and were analyzed for each player and averaged as three different training drills. This information is better clarified in “Locomotor demands” in the “Material and Methods” section.

  1. Lines 116 and 119: Did you registered the exact temperatures and relative humidity for each training session and match? Please, provide this data if you have them. Also, did you control the hydration status of your participants? Since you analyzed performance, hydration status and ambient conditions are important issues to control in order to ensure that your data was not influenced by dehydration (DOI: 10.1007/s40279-017-0738-7)

Response: Unfortunately, we don’t dispose of this information. We agree to the reviewer’s suggestion, and we added to the limitations section the following phrase: “It is also important to acknowledge that once we are analyzing player’s performance, the temperatures and relative humidity for each training session and official match should be registered since hydration status and ambient conditions are important variables that may affect the players' performance”.

Lines 179-185: I am sorry, but I still don't get how you treated your data. How many SSG in how many sessions were monitored for each pitch size? Where they monitored during 10 min or during the whole training session but selected the 10 minutes of maximum effort? This maximum effort was average for the 16 players (all monitored the same time portion) or each player data came from different time portions? Same for the match performance.

Response: We analyzed three different SSG conditions, using three different pitch sizes. Each condition was analyzed on different days. All three SSG conditions lasted 10 minutes and data were averaged for the 16 players all monitored the same time portion. Regarding the match performance, were analyzed the 10 most intense minutes of each player in each of the three matches analyzed.

  1. Lines 189 and 191: Why did you use non-parametric statistics procedures?

Response: We performed non-parametric statistics procedures for two main reasons: (1) the normality of this data were not confirmed; (2) the sample size is small.

  1. Line 204: I am sorry. This is the first time you indicate that the matches were against different level rivals according to classification. I believe that this should have been explained in the methods section. Also, what was the level of the team of the 16 players? Because depending on that, motivation when playing upper o lower level rivals may be different, and thus, performance.

Response: We followed the reviewer’s suggestion and added this information to the “Official matches” in the “Material and Methods” section.

  1. Lines 207-208: I think you mislead this phrase.

Response: We have considered the reviewer’s correction and clarified this information in the manuscript.

  1. Figure 2: It does not look properly. X axis labels are cut.

Response: Response: Figure 1 and figure 2 were revised accordingly.

  1. Lines 236-238: when you say competition, I think you want to say match. I know it means the same, but it is better to unify terms.

Response: We have considered the reviewer’s suggestion and unified the term for “official match” all over the manuscript.

  1. Lines 278 and 280: Again, they may be used indistinctly, but it seems confusing to me when you speak of SSG and LSG as the same concept.

Response: According to the answer given to the first question, we consider the reviewer’s suggestion we revised all text with the purpose of unify the terminology.

  1. Line 293: Please, indicate the main author of this reference (6) for better reading.

Response: We have included the author's name in this reference.

  1. Lines 296-299: I think this affirmation may be logical. However, the conclusion of an umbrella review seems stronger to me. I believe that not always a lower level team plays defensive against an upper level rival. The tactical approach of the staff may influence as well.

Response: We agree with the reviewer on this point. We changed the sentence accordingly to the suggestion.

  1. Line 363: Please, review the references, some of them do not seem to follow the format of the journal.

Response: We revised all the references in the manuscript.